# Collaborative Multiple Players to Address Label Sparsity in Quality Prediction of Batch Processes

**DOI:** 10.3390/s24072073

**Published:** 2024-03-24

**Authors:** Ling Zhao, Zheng Zhang, Jinlin Zhu, Hongchao Wang, Zhenping Xie

**Affiliations:** 1School of Artificial Intelligence and Computer Science, Jiangnan University, Wuxi 214122, China; 6213113153@stu.jiangnan.edu.cn (L.Z.); xiezp@jiangnan.edu.cn (Z.X.); 2Department of Chemical and Biological Engineering, Hong Kong University of Science and Technology, Clear Water Bay, Kowloon, Hong Kong SAR 999077, China; zzhangfj@connect.ust.hk; 3School of Food Science and Technology, Jiangnan University, Wuxi 214122, China; hcwang@jiangnan.edu.cn

**Keywords:** batch processes, soft sensor, co-training, deep learning, semi-supervised learning

## Abstract

For decades, soft sensors have been extensively renowned for their efficiency in real-time tracking of expensive variables for advanced process control. However, despite the diverse efforts lavished on enhancing their models, the issue of label sparsity when modeling the soft sensors has always posed challenges across various processes. In this paper, a fledgling technique, called co-training, is studied for leveraging only a small ratio of labeled data, to hone and formulate a more advantageous framework in soft sensor modeling. Dissimilar to the conventional routine where only two players are employed, we investigate the efficient number of players in batch processes, making a multiple-player learning scheme to assuage the sparsity issue. Meanwhile, a sliding window spanning across both time and batch direction is used to aggregate the samples for prediction, and account for the unique 2D correlations among the general batch process data. Altogether, the forged framework can outperform the other prevalent methods, especially when the ratio of unlabeled data is climbing up, and two case studies are showcased to demonstrate its effectiveness.

## 1. Introduction

Batch processes are now widely used in many high-value-added fields, ranging from food, pharmaceuticals, to the most cutting-edge silicon. The advanced control of these processes is of paramount importance for meeting overall specifications, while requiring a close oversight of certain key performance indicators (KPI) [1,2,3,4,5]. However, variables that are highly influential to them are often expensive to measure, owing to the challenges in sensor installation, analysis delays, and the overall costs [6,7,8]. To alleviate this issue, soft sensor, also known as inferential sensor, has gained popularity in recent decades, and is now recognized as an affordable and reliable prediction method that facilitates the process assessment and optimization [9,10,11,12,13]. 

The advantage of soft sensor lies in its avoidance of using direct hardware to measure variables, rather, using only primary measurements to enable the prediction of quality variables. Depending on the methodology, soft sensors can be divided into first-principles models and data-driven models. Because various statistical and machine learning tools have been readily available for data processing, and the data are more convenient than ever to collect, compared to the previous sophisticated model, data-driven methods are emerging as a predominant solution in the era of big data and artificial intelligence [14,15,16,17,18]. For instance, Gopakumar et al. developed a soft sensor based on deep neural network (DNN), and used it for online estimation of product quality and biomass concentration, showing more significant accuracy in streptokinase and penicillin processes than traditional SVM-based methods [19]. Albeit, with multifarious extensions to boost the model performance, the modeling of soft sensor is uniformly owed to a so-called training phase which fits the relation between the input and output data. This inevitable step generally relies on a substantial amount of data, especially the indispensable Y data, but it is expensive and scarce due to the inherent difficulty in measuring. Therefore, the deficiency of Y data results in the under-labeled issue impedes the ordinary modeling procedure of soft sensors accordingly.

To assuage the issue of label sparsity, semi-supervised learning (SSL) is deemed a feasible strategy during the modeling [20]. Semi-supervised learning uses less labeled data combined with unlabeled data to attain a better performance than using labeled data alone, which has been intensively researched in industrial applications recently [21]. Ge et al. integrated self-training into phase representative PLS models, prompting a better result than traditional PLS models given only a small portion of data was labeled from an injection molding process [22]. Jin et al. proposed an ensemble evolutionary optimization-based pseudo-labeling method (EnEOPL), wherein the Gaussian process regression (GPR) was assigned as base learner, being further enhanced by an ensemble framework to optimize and generate the pseudo-labels [23]. Esche et al. proved that when the time interval between two individual samples is noticeably too large, the SSL effect delivered by proposed deep-kernel learning is significant, as illustrated in Williams-Otto simulation and bioethanol production process [24]. However, these methods only utilized a single base learner, of which the monotone learning characteristic can obscure the real X-Y mapping that could be more perplexed underlying the process, thus easily leading to under-fitting or over-fitting issues during the modeling.

Alternatively, the idea of synthesizing multiple basic learners rather than using just one can enable an array of various perspectives when capturing the features, thereby reducing the variance across different testing datasets or conditions. Li et al. presented a semi-supervised ensemble support vector regression (SSESVR), which inferred and aggregated artificial labels into an under-labeled dataset via more than one base learner to formulate the ensemble hierarchy and facilitate the subsequent collaboration with semi-supervised learning [25]. As another viable route, co-training takes a step towards accentuating the heterogeneity by which the multiple players (base learners) can learn from the dataset [20,26]. The co-training first splits the dataset into two parts which are supposedly independent, then an individual classifier will be assigned to each and trained before screening the unlabeled samples for predicting their pseudo-labels. The pseudo-samples will be later evaluated and selected to replenish the dataset that is irrelevant to the current classifier for the next round of training, until the criteria become useless or the model converges [27]. So far, co-training has found its efficacy in many tasks such as web page classification and has become a fledgling tool in soft sensor applications. To name a few, it could be the co-training regressors (Coreg) that were brought forward by Zhou and Li, where a large amount of unlabeled data dominated the dataset and the routine was used for increasing the regression accuracy [28]. In addition, Bao et al. applied PLS instead of kNN as the basic model, showing better results in simulation and real plant data [29]. Besides, Tang et al. presented a co-training style kernel extreme learning machine (ELM), which deployed two ELMs with designated kernel tricks to label samples in each dataset, demonstrating good capability in handling under-labeled and industrial datasets [30]. However, previous methods based on multiple learners mostly stick to the structure itself, like co-training, without seeking changes in the structure or taking more into account regarding the characteristics of the batch process itself.

The aforementioned work laid a solid foundation in terms of soft sensor modeling and insufficient label treatment. However, there all still several limitations that have not been focused on before: (1) The first are the unique data characteristics brought by the scenario itself. A batch process, wherein the soft sensor is being applied, typically encompasses multiple phases and transitions within and across the batches. This mandates that, not only are some nonlinearities supposed to exist in the process data, but some patterns, known as 2D dynamics, that evolve with time and batch direction would also hamper the data analysis and the subsequent soft sensor modeling, hence vitiating the model capability and its accuracy. (2) Though co-training and the renewed Coreg proved to outperform in sparse label learning (regression), their essences are still based on the original two-learner co-training framework. Regardless of the ensemble nature, if the in-situ base learner is inappropriate or insufficient to capture the heterogeneous features, then a singleton type of base learners may deliver an inferior cooperation when later predicting pseudo-labels, hence degrading the practical significance in multi-phase batch processes. (3) Traditional regression models often either use deep learning for prediction, or directly utilize co-training-based models to predict the final variables. However, in batch process scenarios that lack labeled data, deep learning is limited by the inability to obtain sufficient data labels, and models based on co-training are difficult to capture the complex characteristics of the batch process. Therefore, we believe that only a framework that integrates co-training and deep learning can better solve the problem of quality variable prediction in the current scenario. Therefore, in order to devise a procedure which is geared towards using sparse labels when modeling soft sensors for batch processes, an instructional multi-player framework is showcased in this research to mitigate the scarcity of quality variables and to account for the nonlinearities and dynamics that hinder the modeling and application. The main contributions are thus highlighted as follows:Rather than using only two base learners, we investigate the efficient number of different base learners used in co-training, which paves the way for capturing multi-channeled input features that are leveraged in pseudo-labels generation;Once the pseudo-labels complete the data augmentation, a sliding window is skillfully embedded preceding the feature engineering, to account for the unique 2D dynamics of batch processes;Leveraging the pseudo-labels inferred by the local feature similarity, a deep learning interface named 2D-GSTAE, is further connected to synthesize all the perspectives presented by the previous base learners, promoting a more comprehensive relationship between the input process data and the online estimated output.

Therefore, a framework in the name of co-training with multiple players in deep soft sensor modeling (CO-MP-DSSM) is presented in this work, by which other fields may also benefit given appropriate deep synthesizers and modifications. The framework first adds pseudo-labels to the data through collaborative multi-player learning, then simultaneously extracts two-dimensional dynamic features in the time direction and batch direction through a two-dimensional dynamic sliding window, and finally utilizes deep learning to fuse the features and predict quality variables. The remaining sections of the paper are ordered as follows. In Section 2, a review of the co-training and Coreg is given. In Section 3, the details of the CO-MP-DSSM framework are dissected. In Section 4, case studies on simulation and real fermentation processes are provided to evaluate the prediction performance. At the end, the summary and outlook of soft sensor in batch processes are given in the last section.

## 2. Preliminary

### 2.1. Original Co-Training

The concept of co-training is originally proposed by Blum and Mitchel [27]. Its implementation is shown in Figure 1, where L is labeled dataset, $L_{1}$ and $L_{2}$ are the different views of training set, $h_{1}$ and $h_{2}$ are the different classifier, U donates the unlabeled dataset and U’ represents the random selected unlabeled dataset, t is the test dataset and y’ represents the prediction result. Co-training firstly divides the labeled dataset into two supposedly independent and redundant views, $L_{1}$ and $L_{2}$, then trains different classifiers $h_{1}$ and $h_{2}$ on them and utilizes the prediction results to expand each other’s training set. The reduced part of U’ will be supplemented from U. After k times of the above iterative training, $h_{1}$ and $h_{2}$ are finally used to classify the test set. Co-training has proven useful in classification scenarios with few labeled samples, but due to the aforementioned limits, it can be still inferior when directly used in soft sensor modeling. 

### 2.2. Co-Training Regressors (Coreg)

As an effective semi-supervised learning paradigm, co-training has received a lot of attention since its novel concept, but related research is only based on classification problems. Therefore, Zhou and Li proposed a co-training style regression model, co-training regressors (Coreg) [28]. Since it is very similar to co-training, the structure of the Coreg algorithm is also shown in Figure 1 and the difference is marked as red, where $r_{1}$ and $r_{2}$ are changed to the different regressors. Both $r_{1}$ and $r_{2}$ are based on kNN regressor but use different Minkowski distances. The Minkowski distance is calculated as follows,
(1)$$Minkowsky_{p}(x_{a},x_{b})={\left(\sum_{i=1}^{d}{\left|x_{ai}-x_{bi}\right|}^{p}\right)}^{\frac{1}{p}}$$
where $x_{a}$ and $x_{b}$ are two samples in the multi-dimensional space, d represents the total dimension of the sample and p denote the distance order. Minkowski distance is a metric only when p is not less than 1. Generally speaking, when the value p is large the distance metric is more sensitive to samples, and when the value p is small, the distance metric is more robust. 

During the training phase, Coreg algorithm first duplicates the labeled dataset *L* into two copies, $L_{1}$ and $L_{2}$, respectively, as training sets for the two regressors $r_{1}$ and $r_{2}$ to initialize them. Coreg then works in a similar way to co-training, but the specific steps are different. When it iterates, each time the two regressors first traverse all the data in U’, predict the label of unlabeled data $U_{j}$, and add it to the training set before retraining the regressors. Then the regressor is used to predict the k nearest neighbors to $U_{j}$ from the labeled dataset, and the loss is calculated as follows:(2)$$U_{j}=\underset{U_{j}\in U’}{\mathrm{Argmax}}(\sum_{x_{i}\in \Omega }{\left(y_{i}-r_{a}(x_{i})\right)}^{2}/\left|\Omega \right|-\sum_{x_{i}\in \Omega }{\left(y_{i}-r_{a}’(x_{i})\right)}^{2}/\left|\Omega \right|)$$
where $r_{a}$ and $r_{a}’$ are one of the regressors and the retrained regressor after adding the predicted label, and Ω is the k nearest neighbors to $U_{j}$ in the labeled dataset. The $U_{j}$ with maximum positive loss and its predicted label will be added to another labeled dataset, while no data will be added if without positive loss. If neither $L_{1}$ or $L_{2}$ change, the iteration will end early. Finally, $r_{1}$ and $r_{2}$ will be retrained using the changed $L_{1}$ and $L_{2}$. After training is completed, Coreg predicts the test set t as follows:(3)$$y’=(r_{1}(x)+r_{2}(x))/2$$

## 3. Proposed Method

In this paper, in order to fully utilize the small amount of labeled data as well as the large amount of unlabeled data, and predict the quality variables of the batch processes, we propose a co-training-based semi-supervised deep learning model CO-MP-DSSM. The structure of CO-MP-DSSM is shown in Figure 2, with the details presented explicitly as follows. Suppose the process data and quality data of batch processes have been collected and organized as $X\left(I\times J\times K\right)$ and $Y\left(I\times 1\times L\right)$, where I indicates the number of batches, J represents the number of variables, K is the number of sampling numbers for process variables, and L denotes the sampling number for quality variable. In CO-MP-DSSM, the data are processed and modeled as follows:

### 3.1. Collaborative Multiple-Player Structure to Infer the Labels in Individual Perspectives

Based on previous Coreg, we proposed a collaborative scheme with multiple players to mitigate the scarcity of the labeled data. The pseudocode for this algorithm is shown as Algorithm 1. Regarding this, we firstly divide the batch processes data into labeled data and unlabeled data and copy the labeled data into several copies, corresponding to the predefined number of regressors.
Algorithm 1 Collaborative Multiple Players LearningInput: labeled dataset L, unlabeled dataset U,  number of the nearest neighbors k,

the maximum number of iteratios N,

$\text{distance}\;\text{order}\;p_{1},p_{2},\ldots ,p_{n}$Procedure:
$L_{1}\leftarrow L;\;L_{2}\leftarrow L;\;\ldots ;\;L_{n}\leftarrow L;\;\text{number}\;\text{of}\;\text{iterations}\;num\leftarrow 0$
$P_{1}\leftarrow kNN(L_{1},k,p_{1});\;P_{2}\leftarrow kNN(L_{2},k,p_{2});\ldots P_{n}\leftarrow kNN(L_{n},k,p_{n})\;\;\;\;\;\;\;\;\;\#\;\mathrm{Initialize}\;\mathrm{all}\;\mathrm{the}\;\mathrm{datasets}\;\mathrm{and}\;\mathrm{regressors}$
while num<N do

num←num+1

for j∈{1,2,…,n}do                        # Perform iteration over each regressor


for each x∈U do                        # Perform iteration over each unlabeled data



$\hat{y}\leftarrow r_{j}(x)\;\;\;\;\;\;\;\;\;\;\;\;\;\;\;\;\;\;\;\;\;\;\;\;\;\;\;\;\;\;\;\;\;\#\;\mathrm{Predict}\;\mathrm{the}\;\mathrm{pseudo-label}\;\mathrm{for}\;x$



$P_{j}’\leftarrow kNN(\{(x,\hat{y})\}\cup L_{j},k,p_{j})\;\;\;\;\;\;\;\;\;\;\;\;\;\;\;\#\;\mathrm{Train}\;\mathrm{new}\;\mathrm{regressor}\;\mathrm{using}\;\mathrm{data}\;\mathrm{containing}\;x\;\mathrm{and}\;\hat{y}$



$a\leftarrow GetNeighbors(x,k,L_{j})\;\;\;\;\;\;\;\;\;\;\;\;\;\;\;\;\;\;\;\;\#\;\mathrm{Get}\;\mathrm{the}\;\mathrm{neighbors}\;\mathrm{of}\;x\;\mathrm{from}\;\mathrm{labeled}\;\mathrm{data}$



$\Delta_{x_{i}}\leftarrow \sum_{x_{i}\in a}((y_{i}-P_{j}’{\left(x_{i}\right)}^{2}-(y_{i}-P_{j}{\left(x_{i}\right)}^{2})/|a|\;\;\;\;\;\;\;\;\;\;\;\#\;\mathrm{Calculate}\;\mathrm{the}\;\mathrm{difference}\;\mathrm{before}\;\mathrm{and}\;\mathrm{after}\;\mathrm{adding}$


end for


$if\;exist\;\Delta_{x_{i}}<0\;then$



$x_{i}\leftarrow \underset{x_{i}\in U}{\mathrm{Argmin}}\Delta_{x_{i}};{\hat{y}}_{i}\leftarrow P_{j}(x_{i})\;\;\;\;\;\;\;\;\;\;\;\;\;\;\;\;\;\;\#\;\mathrm{Select}\;\mathrm{the}\;\mathrm{optimal}\;x\;\mathrm{and}\;\hat{y}\;\mathrm{from}\;\mathrm{unlabeled}\;\mathrm{data}$



$\pi_{j}\leftarrow \{(x_{i},{\hat{y}}_{i})\},U\leftarrow U-\pi_{j}\;\;\;\;\;\;\;\;\;\;\;\;\;\;\;\;\;\;\;\;\;\;\#\;\mathrm{Remove}\;x\;\mathrm{from}\;\mathrm{the}\;\mathrm{unlabeled}\;\mathrm{data}$


else π←∅


end if

end for

$\pi_{s}\leftarrow \pi_{1}\cup \pi_{2}\cup \ldots \cup \pi_{n}$

$L_{1}\leftarrow L_{1}\cup (\pi_{s}-\pi_{1});L_{2}\leftarrow L_{2}\cup (\pi_{s}-\pi_{2});\ldots ;L_{n}\leftarrow L_{n}\cup (\pi_{s}-\pi_{n})\;\#\;\mathrm{Add}\;\mathrm{the}\;\mathrm{optimal}\;x\;\mathrm{and}\;\hat{y}\;\mathrm{to}\;\mathrm{each}\;\mathrm{other}\;\mathrm{labeled}\;\mathrm{data}$

$if\;neither\;of\;L_{1},L_{2},\ldots ,L_{n}\;change\;then\;exit$

else


$P_{1}\leftarrow kNN(L_{1},k,p_{1});P_{2}\leftarrow kNN(L_{2},k,p_{2});\ldots$


$P_{n}\leftarrow kNN(L_{n},k,p_{n})\;\;\;\;\;\;\;\;\;\;\;\;\;\;\;\;\;\;\;\;\;\;\#\;\mathrm{Retrain}\;\mathrm{the}\;\mathrm{regressors}\;\mathrm{with}\;\mathrm{updated}\;\mathrm{data}$

end if
end whileend Procedure$Output:r(x)\leftarrow (r_{1}(x)+r_{2}(x)+\ldots +r_{n}(x))/n\;\mathrm{is}\;\mathrm{used}\;\mathrm{to}\;\mathrm{predict}\;\mathrm{the}\;\mathrm{pseudo}\;\mathrm{label}\;\mathrm{of}\;\mathrm{test}\;\mathrm{set}\;t$

After initializing the regressors using labeled data, we iteratively train the regressors. For each iteration, we train each regressor in turns, using them to traverse all data in the unlabeled dataset. For each unlabeled datum, we first predict its pseudo-label through the previous regressor P, and retrain it using the dataset after adding the data and its pseudo-label to obtain the regressor P’. In order to evaluate the performance of the regressor after adding the data and its pseudo-labels, we approximately take the k nearest neighbors of the data in the labeled data and compare the RMSE predicted by P and P’. The loss of the regressor changes as follows:(4)$$\hat{x}=\underset{\hat{x}\in U}{\mathrm{Argmin}}(\sum_{x_{i}\in a}{\left(y_{i}-P’(x_{i})\right)}^{2}-\sum_{x_{i}\in a}{\left(y_{i}-P(x_{i})\right)}^{2}/\left|a\right|)$$
where a denotes the k nearest neighbors of the data $\hat{x}$. According to previous research, we believe that the smaller the loss, the better the effect of adding this data to improve the prediction performance of the regressor. For this training, we obtain the unlabeled data corresponding to the smallest negative loss and its pseudo-label and add it to the training set of other regressors. If no loss is negative, we believe that no data can improve the performance of the regressor and we do not add any data to other training sets. Iterative training is repeated until the number of times reaches the upper limit or all regressors are unable to select unlabeled data that can improve the prediction effect. After training is complete, we use unlabeled training data as input and predict their labels. The label of each datum is calculated as:(5)$$y’=(P_{1}(x)+P_{2}(x)+\ldots +P_{n}(x))/n$$
where $P_{1}$, $P_{2}$, …, $P_{n}$ represent the regressors trained using different Minkowski distances. Finally, the training set is reshaped so that it has the same shape as at the beginning.

### 3.2. Preprocess and 2D Slide Window Preceding the Modeling

To reduce the impact on the model caused by differences in data values and units, for train x, train y, and test x we standardize the three-dimensional data. For instance, the data with shape I×J×K is first transformed into J×I×K. Next, we divide the data into a matrix of dimension $J\times \left(I\times K\right)$ and standardize it using z-score by row, respectively. For each matrix, the standard deviation is:(6)$$\sigma =\sqrt{\sum_{i=1}^{N}{\left(x_{i}-\bar{x}\right)}^{2}/N}$$
where N is the number of elements in the matrix, and $\bar{x}$ represents the mean of the elements. In the matrix each element x is normalized as:(7)$$s=(x-\bar{x})/\sigma$$
where s denotes the result of standardizing. Afterwards, the matrix is transformed back into I×J×K.

In order to extract two-dimensional dynamic features from time and batch directions simultaneously, we utilize 2D sliding windows to process the data. For train x, train y and test x, we use a 2D sliding window to first slide in the time direction moment by moment until the end, then slide to the next position in the batch direction and slide in the time direction from the beginning. The details of the 2D sliding window method are illustrated in Figure 3, where the data undergo dimensionality reduction after being processed through the sliding window. For example, if the data of I×J×K is slid using the sliding window of p×q, we obtain data with shape of $p\times q\times J\times \left(I-p+1\right)\times \left(K-q+1\right)$, which is a five-dimensional tensor. The size of the sliding window is selected according to the data dimension and can be adjusted according to the experimental results. Since tensors with too high dimensions cannot be trained using deep neural networks, and too many dimensions are meaningless, we first spliced the data in the sliding window batch by batch, and then arranged the sliding windows in sliding order to obtain processed data $\left(p\times q)\times J\times \;[\left(I-p+1\right)\times \left(K-q+1\right)\right]$.

### 3.3. Deep Learning: Fusing and Prediction

According to our previous research, GSTAE achieves better results for batch process quality variable prediction compared to traditional SAE for quality prediction in batch processes [31]. Compared to traditional SAE, GSTAE not only utilizes gate unit to integrate the feature of data in different layers, but also adds quality-related information in the pre-training stage to allow the network to extract more features related to quality variables [32]. The pseudocode of GSTAE is shown in Appendix A. So, in the end, the processed data is used to train GSTAE model, whose training phase consists of supervised pre-training stage and fine-tuning stage. In the pre-training stage, GSTAE first attempts to extract more abstract features by mapping the input data to a hidden layer with fewer neurons, and then reconstructs the input through the hidden layer to reduce the loss of information. GSTAE reconstructs the data as:(8)$${\left[\tilde{x},\tilde{y}\right]}^{T}=\tilde{f}(W_{2}f(W_{1}x+b_{1})+b_{2})$$
where x is input matrix, $\tilde{x}$ and $\tilde{y}$ are the reconstruction results of x, and y, $W_{1}$ and $W_{2}$ are the weight matrix, $b_{1}$ and $b_{2}$ are the bias vector. The network is trained layer-by-layer in the above order. The number of neurons in the first layer of the network is equal to the number of input variables, and then the number of neurons in each layer gradually decreases. The prediction of quality variables in each layer can avoid the loss of information related to quality variables while obtaining more abstract process variable characteristics. In the fine-tuning stage, the model predicts the label and tries to fit the real label, so the loss is calculated as
(9)$$Loss=\sum_{i=1}^{N}{\left\|y_{i}-{\tilde{y}}_{i}\right\|}^{2}$$
where N is the total amount of data. After training is completed, the network inputs sample features and weights the predicted labels of each layer through the gate unit. The gate unit controls the impact of this layer of information on the final result through the weight matrix, bias vector and activation function. Then the predictions of each layer are added in sum to obtain the final predicted label. The label is calculated as
(10)$$\tilde{y}=\sum_{i=1}^{n}\sigma (W_{gi}\times h_{i}+b_{gi})⊗\tanh(W_{oi}\times h_{i}+b_{oi})$$
where n represents the layer of GSTAE, $W_{g}$ and $W_{o}$ are the weight matrix of the gate, $b_{g}$ and $b_{o}$ are the bias vector of the gate.

### 3.4. Evaluation Indicators

In order to facilitate the comparison of prediction effects between models, we choose the most widely used performance evaluation indices RMSE and R^2^ as indicators of model prediction effects. The RMSE and R^2^ are calculated as
(11)$$RMSE=\sqrt{\sum_{i=1}^{n}{\left(y_{i}-{\tilde{y}}_{i}\right)}^{2}/n}$$
(12)$$R^{2}=\sum_{i=1}^{n}{\left({\tilde{y}}_{i}-\bar{y}\right)}^{2}/\sum_{i=1}^{n}{\left(y_{i}-\bar{y}\right)}^{2}$$

In general, a decrease in RMSE and an increase in R^2^ indicate an improved predictive performance of the model.

## 4. Case Study

### 4.1. Penicillin Fermentation Simulation Case

Penicillin is one of the most commonly used antibiotics. Most of it is synthesized through fermentation and purification. The fermentation process of penicillin is a typical fed-batch process, which has strong nonlinearity and two-dimensional dynamic characteristics. To prove the effectiveness of the model, we utilize penicillin fermentation simulation platform PenSim V2.0 to generate simulated data [33]. Since the data generated by PenSim V2.0 are close to the real situation, and most research in related fields also uses this platform, we also use this platform for simulation experiments.

#### 4.1.1. Experiment Design in Simulation Case

By using the simulation platform, we generated a total of 100 batches of data, of which 75 batches were used as training sets and 25 batches were used as test sets. The total fermentation time is set to 400 h, and the sampling interval is set to 1 h. In order to simulate the two-dimensional dynamic features in real process, we set initial conditions as follows. For batch direction, pH is set to gradually grow from 4.95 to 5.2, temperature is set to gradually grow from 297.5 to 298.5. Besides, substrate concentration is set to 15, agitator power is set to 30, carbon dioxide concentration is set to 0.5, aeration rate is set to 8.6 and culture volume is set to 100. The above variables of different batches fluctuate within about 1% of the set value. The remaining variables are set to the default values of Pensim V2.0. The variables selected for modeling are displayed in Table 1. Because the data generated by the Pensim is too ideal, we add a certain amount of noise to the data to make it closer to the real result.

#### 4.1.2. Parameter Settings in Simulation Case

To showcase the superiority and evaluate the performance of models, SAE, SIAE, GSTAE, 2D-GSTAE, SS-GSTAE, SS-2D-GSTAE, Coreg, and CO-MP-DSSM are constructed. Among them, 2D-GSTAE is the CO-MP-DSSM without a collaborative multiple-player structure, Coreg is the collaborative multiple-player structure without the other part, and SS-2D-GSTAE is 2D-GSTAE combined with traditional semi-supervised learning structurally. The parameters of the model and comparison models are set as follows. The structure of CO-MP-DSSM is set as {11, 9, 7, 4}, where 11 is the number of neurons in the first layer must be consistent with the number of variables, and 9, 7, 4 are the numbers of neurons in hidden layers. According to the test, the structure above has given the best prediction performance among all structures. The learning rate of the model is set as 0.01. The trade-off coefficient λ is set to 0.5. For comparison models, the structure of SAE, SIAE, GSTAE, SS-GSTAE, 2D-GSTAE are also set as {11, 9, 7, 4}. Such identical parameter settings not only make all models perform well, but also make comparisons between models relatively fair. The learning rate of SAE, SIAE, GSTAE, SS-GSTAE, 2D-GSTAE, SS-2D-GSTAE is set as 0.01. The trade-off coefficient λ of GSTAE, SS-GSTAE, 2D-GSTAE, SS-2D-GSTAE is set to 0.5. The number of regressors in co-training is set to 2 for Coreg and 3 for CO-MP-DSSM. The sizes of the two-dimensional sliding windows of 2D-GSTAE, SS-2D-GSTAE, and CO-MP-DSSM in batch and time are set to 7 and 5. The number of nearest neighbors of kNN in Coreg and CO-MP-DSSM is set to 15.

#### 4.1.3. Results and Analysis in Simulation Case

The results of models are presented in Table 2, where the proposed model and best performance in each ratio is highlighted in bold. The summary of comparative results among different methods is displayed in Figure 4 with standout methods represented in different colors. All networks were initialized with ten different random seeds, and their result indicators were averaged to obtain the final experimental results. The sampling ratio in Table 2 denotes the proportion of labeled data in total data. Since this case is a simulation experiment, the effect prediction of the model with small noise and other interference factors is relatively stable, thereby facilitating a clearer comparison of the results influenced by the model’s structure. It is not difficult to find from the charts and tables that traditional methods such as SAE and SIAE are not fitted well for batch processes, while the model based on GSTAE achieves significantly better results. Since SAE and SIAE only consider x-related information in the pre-training stage, the models’ ability to extract quality-related information is limited. Moreover, the simple structure further limits the model’s nonlinear feature extraction capabilities, resulting in poor prediction performance. Therefore, it is evident that the structure of the basic model determines the prediction effect of the model to a certain extent.

The remaining models achieve similar results when ample labeled data is available, but the gap of effects gradually appear as the labeled data are reduced. When labeled data are scarce, the effects of GSTAE and 2D-GSTAE, experience the most pronounced deterioration in performance. This is because the model that is structured as supervised learning cannot effectively utilize unlabeled data, leading to significant performance degradation as the quantity of labeled data diminishes. Compared with the former, SS-GSTAE and SS-2D-GSTAE utilize semi-supervised learning to deal with unlabeled data and achieve certain results in dealing with data with labeled data accounting for more than 1/25. In order to highlight the effect of SS-2D-GSTAE, we mark its results with a blue line in Figure 4. However, since labeled data may be scarcer in actual situations, the models’ effect of semi-supervised learning will also decline significantly once it exceeds a specific threshold. In addition, no matter at which ratio, the model with a 2D sliding window consistently outperform those without, due to their enhanced ability to capture the dynamic characteristics of batch processes.

However, when labeled data are scarce, the situation is different. Compared with other semi-supervised learning methods, the models based on co-training emphasize the heterogeneity of the structural design. So when faced with a large amount of labeled data the effect is close to traditional semi-supervised learning based method, and it can avoid underfitting and overfitting to achieve better performance when the labels are scarce. Among them, because CO-MP-DSSM integrates multi-player co-training, 2D dynamic feature extraction and deep learning model learning, it can not only obtain more trustworthy pseudo-labels, but also avoid potential issues caused by the too simple structure of the base model by using deep learning. Therefore, CO-MP-DSSM achieves the smallest RMSE and the biggest R^2^ among all the models. In order to highlight the effect of CO-MP-DSSM, we mark its results with a red line in Figure 4.

### 4.2. Real Lactic Acid Bacteria Fermentation Case

In order to further verify the effectiveness of the model, we conducted real lactic acid bacteria fermentation experiments. Lactobacillus plantarum is a member of the genus Lactobacillus and has the potential to be widely used in the food and fermentation industries. It has antioxidant, antibacterial and antiviral potential, helps maintain the balance of intestinal flora, and promotes the health of the digestive system. To evaluate the effectiveness of the model, we selected Lactobacillus plantarum HuNHHMY71L1 as the fermentation strain.

#### 4.2.1. Experiment Design in Real Case

In this real case, we obtained a total of 20 batches of data through fermentation, of which 15 batches were used as training sets and 5 batches were used as test sets. According to previous reference, the total fermentation time was set to 8 h, the sampling interval of process variables was set to 1 min, while due to conditional restrictions the sampling interval of quality variable was set to 30 min. Supplementary culture medium was added to the fermenter after 2 h of fermentation. In addition, MRS was used as the culture medium, the fermentation temperature was set to 37 °C allowing a margin of error of 0.2, the pH was set to 6 allowing a margin of error of 0.02, the culture volume was set to 3 L, and the stirring speed is was to 100 rmp. Due to changes in external conditions during the fermentation process and operational errors during the experiment, the fermentation process had two-dimensional dynamic characteristics like other fed-batch processes. Figure 5 depicts the instruments utilized in our fermentation process, along with corresponding schematic diagrams of their specific structures. The measured variables are presented in Table 3.

#### 4.2.2. Parameter Settings in Real Case

To compare the performances of the models, the same models as in the previous case were constructed, and some hyperparameters were adjusted according to the data. The parameters of the model and comparison models were set as follows. The structure of models was set as {7, 6, 5, 3}, where 7 is the number of neurons in the first layer that must be consistent with the number of variables, and 6, 5, 3 are the number of neurons in hidden layers. The number of regressors in co-training was set to 2 for Coreg and 3 for CO-MP-DSSM. The sizes of the two-dimensional sliding windows of 2D-GSTAE, SS-2D-GSTAE and CO-MP-DSSM in batch and time were set to 3 and 3. The number of nearest neighbors of kNN in Coreg and CO-MP-DSSM is set to 10.

#### 4.2.3. Result and Analysis in Real Case

The results of models are shown in Table 4, where the proposed model and best performance in each ratio is highlighted in bold. Figure 6 employs boxplots to compare the predictive performance of different models and the stability of their predictions. The sampling ratio in Table 4 denotes the proportion of labeled data in total data. Due to the limitations of sampling in the actual process, only approximately 1/30 of the total dataset could be labeled. Therefore, we can only compare the performance of various models when labels are scarce. We have specifically chosen to analyze results for three different sampling ratios as further reduction in labeled data would render the experiment less meaningful. Moreover, variations in the intervals of labeled data could compromise the experimental design. Additionally, it is important to note that the effect of the model may exhibit relative instability due to the significant influence of external factors in the actual operational process.

Despite the limited amount of labeled data, the quality-related information cannot be ignored for the final prediction result, resulting in the suboptimal performance of SAE and SIAE. GSTAE has demonstrated stable performance owing to its structural advantages, while models SS-GSTAE and 2D-GSTAE have achieved better results through the incorporation of semi-supervised and 2D sliding window structures, respectively. However, unlike the previous case, SS-2D-GSTAE did not achieve better results than SS-GSTAE and 2D-GSTAE. This may be due to the smaller number of batches limiting the size selection of the sliding window, and the smaller number of labeled data resulting in a lack of reliable label information in some windows. Therefore, when dealing with actual cases, effective combination of model structures is more important than simple stacking structures. Since only a simple base learner is used and the strong nonlinearity and 2D dynamic characteristics of the batch processes are not taken into account, sole reliance on Coreg does not yield satisfactory outcomes. Through the effective combination of multi-player co-training, 2D dynamic sliding windows and deep learning models, CO-MP-DSSM leverages their strengths while mitigating weaknesses, thereby effectively learning the characteristics of the batch process and achieving the smallest RMSE and biggest R^2^ among all the models. In order to highlight the effect of CO-MP-DSSM, we mark its results with an orange boxplot in Figure 6. 

Figure 7 and Table 5 show another reason behind the superior performance of CO-MP-DSSM over Coreg, and provide another perspective on why we chose to use three regressors in this case. When we used CO-MP-DSSM with 1–6 regressors for prediction tasks, we observed that the model utilizing three regressors not only outperformed other configurations but also exhibited greater stability in prediction outcomes across most scenarios. Furthermore, in most cases when the number of regressors is less than the best number, increasing the number of regressors will improve the model effect, while when the number of regressors is greater than the best number, decreasing the number of regressors will improve the model performance. Compared with primitive 2D-GSTAE, CO-MP-DSSM using three regressors is not only stable, but also performs better in most cases, especially when the number of labeled data is relatively trustworthy. In addition to the accuracy brought by multiple players, we believe that even using less accurate pseudo-labels can also improve the prediction effect of the model, so CO-MP-DSSM achieves far better results than 2D-GSTAE. So, in this article we used CO-MP-DSSM with three players for training.

## 5. Conclusions

Label scarcity is an actual problem faced by many deep-learning-based methods when working on chemical soft sensor modeling. In this paper, a collaborative multiple-player combined framework, named CO-MP-DSSM, has been put forward to address the label sparsity in the quality prediction of batch processes. First, dissimilar to the conventional scheme of co-training where there used to be only two players, the number of effective players has been studied to generate more significant pseudo labels, even when the ratio of true labels is very scarce. Afterwards, a 2D sliding window is used on the batch data to capture the correlations in both the time and batch directions. Finally, for illustration, the processed data is put into the GSTAE model for the up-scale extraction of features and quality prediction. Experimental results on two datasets demonstrate that CO-MP-DSSM achieves better results than other models without this technique. However, since each of the multi-players is supposed to use kNN to iterate over all the unlabeled data, the computation is relatively expensive in the training phase compared with other methods. In the future, researchers may explore more efficient perspectives for players to measure the similarity between the data, aiming to refine methods that can achieve better results while minimizing computational resources.

## Figures and Tables

**Figure 1 sensors-24-02073-f001:**
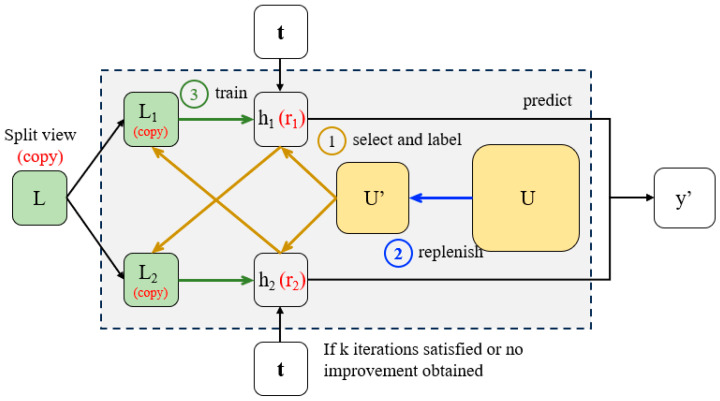
Dissected diagram of co-training and Coreg (red) algorithm. (The major difference lies in that co-training has its dataset split by two views from the raw data, while Coreg simply duplicates the data into two copies. Besides, the base learners of them are termed classifiers and regressors, respectively).

**Figure 2 sensors-24-02073-f002:**
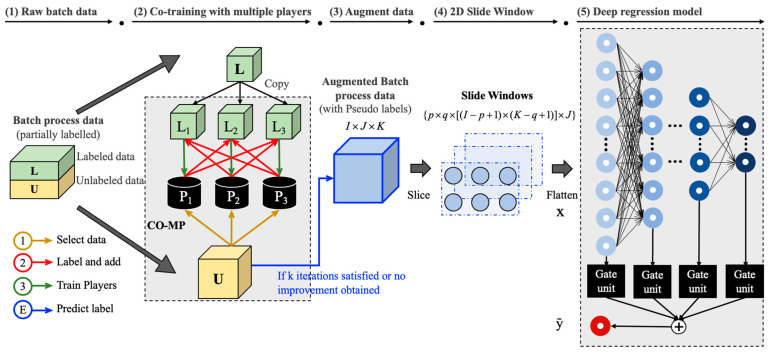
Procedure of proposed CO-MP-DSSM framework: (1) details explained in Section 3; (2) & (3) Section 3.1; (4) Section 3.2; (5) Section 3.3.

**Figure 3 sensors-24-02073-f003:**
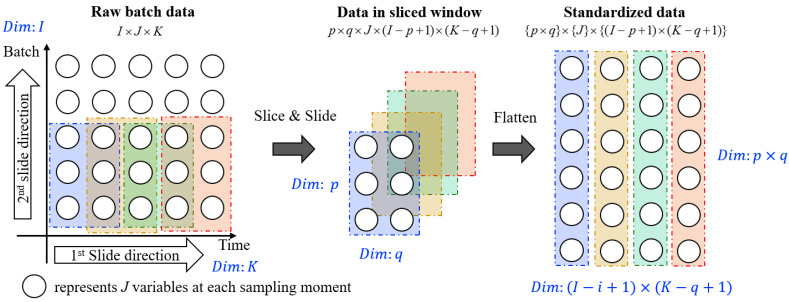
Flowchart of the employed 2D slide window (As depicted above, the window first slides forward along the time direction, then shifts into the beginning of the next batch once the current batch ends.)

**Figure 4 sensors-24-02073-f004:**
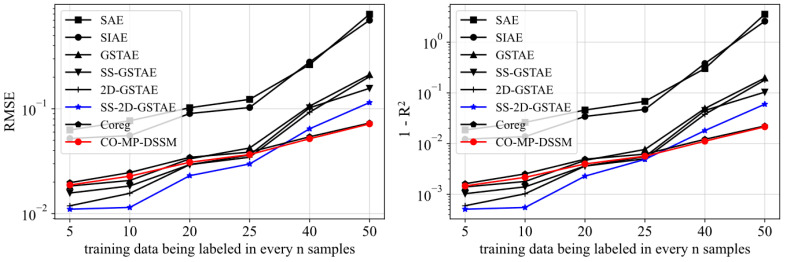
Aggregated results of average RMSE and R^2^ of models compared in the case of Pensim dataset.

**Figure 5 sensors-24-02073-f005:**
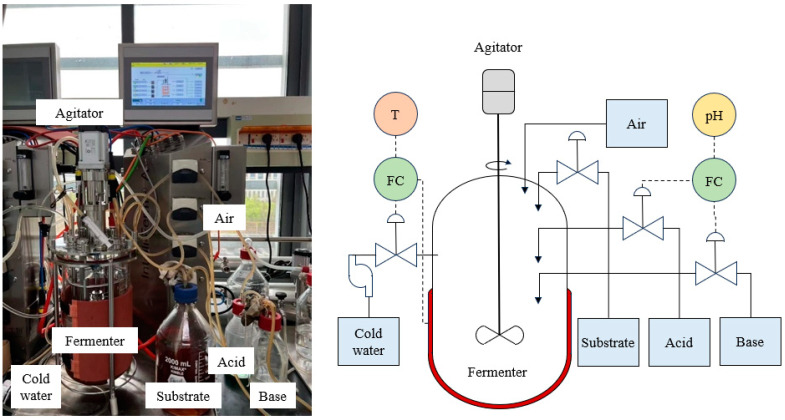
Instruments for lactic acid bacteria fermentation (the left side is the actual picture, and the right side is the schematic diagram. The equipment on the picture corresponds to each other).

**Figure 6 sensors-24-02073-f006:**
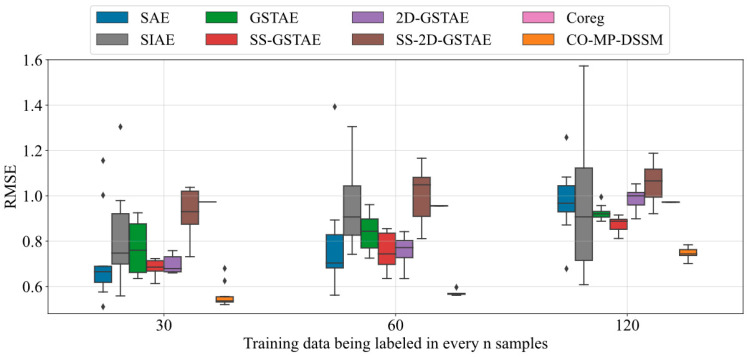
Comparative performances (boxplot) of RMSE in real lactic acid bacteria fermentation dataset.

**Figure 7 sensors-24-02073-f007:**
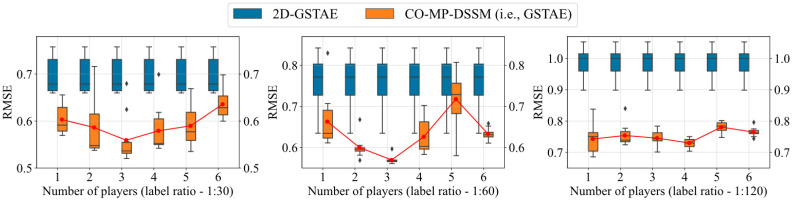
Comparative performances (boxplot) of RMSE from models with various players and label ratios, based on the real lactic acid bacteria fermentation dataset. (When number of players is set to 1, 2D-GSTAE is structurally equal to CO-MP-DSSM, except for slight higher RMSE due to no pseudo labels involved in its training. For better illustration, we repeat the plot for 2D-GSTAE across different situations, to highlight the accuracy and stability improvements by the proposed method.)

**Table 1 sensors-24-02073-t001:** Selected variables of penicillin fermentation process.

Number	Variable	Unit
1	aeration rate	L/h
2	agitator power	W
3	substrate feed rate	L/h
4	substrate feed temperature	K
5	substrate concentration	g/L
6	culture volume	L
7	carbon dioxide concentration	g/L
8	pH	-
9	temperature	K
10	generated heat	kcal
11	cold water flow rate	L/h
y	penicillin concentration	g/L

**Table 2 sensors-24-02073-t002:** Tabulated average RMSE and R^2^ of models compared in penicillin fermentation process. (The model in bold is the method proposed in this article, and the number in bold is the best one in the column of data)

Method	5:1		10:1		20:1		25:1		40:1		50:1	
	RMSE	R^2^	RMSE	R^2^	RMSE	R^2^	RMSE	R^2^	RMSE	R^2^	RMSE	R^2^
SAE	0.0628	0.9814	0.0769	0.9737	0.1024	0.9541	0.1230	0.9319	0.2637	0.6980	0.7977	−2.5510
SIAE	0.0521	0.9878	0.0555	0.9862	0.0902	0.9657	0.1029	0.9528	0.2791	0.6203	0.6989	−1.5870
GSTAE	0.0183	0.9986	0.0207	0.9982	0.0333	0.9953	0.0423	0.9923	0.1060	0.9507	0.2119	0.8036
SS-GSTAE	0.0157	0.9990	0.0184	0.9986	0.0293	0.9964	0.0363	0.9945	0.1023	0.9549	0.1568	0.8964
2D-GSTAE	0.0119	0.9994	0.0156	0.9990	0.0293	0.9964	0.0345	0.9950	0.0927	0.9622	0.2027	0.8202
SS-2D-GSTAE	**0.0110**	**0.9995**	**0.0115**	**0.9994**	**0.0230**	**0.9977**	**0.0298**	**0.9951**	0.0645	0.9819	0.1147	0.9402
Coreg	0.0197	0.9984	0.0247	0.9975	0.0344	0.9951	0.0386	0.9938	0.0540	0.9878	0.0732	0.9777
**CO-MP-DSSM**	0.0188	0.9985	0.0228	0.9978	0.0309	0.9960	0.0366	0.9943	**0.0518**	**0.9888**	**0.0718**	**0.9786**

**Table 3 sensors-24-02073-t003:** Selected variables of lactic acid bacteria fermentation process.

Number	Variable	Unit
1	temperature	K
2	pH	-
3	dissolved oxygen	-
4	agitator rate	r/min
5	acid supplement	mL
6	base supplement	mL
7	substrate supplement	mL
y	lactic acid bacteria concentration	-

**Table 4 sensors-24-02073-t004:** Tabulated average RMSE and R^2^ of models compared in lactic acid bacteria fermen-tation process. (The model in bold is the method proposed in this article, and the number in bold is the best one in the column of data)

Method	30:1		60:1		120:1	
	RMSE	R^2^	RMSE	R^2^	RMSE	R^2^
SAE	0.7204	0.9186	0.7804	0.9031	0.9759	0.8574
SIAE	0.8215	0.8951	0.9429	0.8657	0.9597	0.8513
GSTAE	0.7704	0.9111	0.8365	0.8965	0.9242	0.8746
SS-GSTAE	0.6832	0.9314	0.7515	0.9163	0.8734	0.8880
2D-GSTAE	0.6962	0.9287	0.7622	0.9143	0.9843	0.8576
SS-2D-GSTAE	0.9266	0.8727	1.0061	0.8496	1.0604	0.8341
Coreg	0.9731	0.8611	0.9558	0.8660	0.9722	0.8614
**CO-MP-DSSM**	**0.5593**	**0.9538**	**0.5698**	**0.9524**	**0.7455**	**0.9184**

**Table 5 sensors-24-02073-t005:** The average prediction RMSE and R^2^ of co-training methods with different number of players in lactic acid bacteria fermentation process. (The model in bold is the method proposed in this article, and the number in bold is the best one in the column of data)

Number of Players	30:1		60:1		120:1	
	RMSE	R^2^	RMSE	R^2^	RMSE	R^2^
1	0.6032	0.9465	0.6630	0.9349	0.7427	0.9188
2	0.5863	0.9490	0.6000	0.9471	0.7540	0.9165
**3**	**0.5593**	**0.9538**	**0.5698**	**0.9524**	0.7455	0.9184
4	0.5792	0.9505	0.6261	0.9422	**0.7301**	**0.9218**
5	0.5897	0.9487	0.7177	0.9239	0.7807	0.9106
6	0.6359	0.9406	0.6329	0.9412	0.7649	0.9142

## Data Availability

Data are contained within the article.

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
