# Peer review of "Collaborative Multiple Players to Address Label Sparsity in Quality Prediction of Batch Processes"

_sensors, 2024, doi:10.3390/s24072073_

Round 1
Reviewer 1 Report
Comments and Suggestions for Authors
This paper proposes a fledgling technique called co-training for leveraging only a small ratio of labeled data, to hone and formulate a more advantageous framework in soft sensor modeling. Specific concerns are given below.
1. The authors introduced many related work in Introduction part, which will drown out the main essence of the paper. The reviewer suggests the authors reorganize the Introduction. Some unimportant literature can be placed in related work part.
2. Although the authors concluded several limitations of previous work, the motivation of the proposed method still needs to be enhanced.
3. It's better to show the main steps of the proposed CO-MP-DSSM method in Introduction part.
4. Formulas and symbols should be written beautifully.
5. Algorithm 1 is not a style of pseudo-code. Currently, the Algorithm can be simplified. In addition, some annotations should be added to the Algorithm.
6. It's better to add reference labels to comparison methods in Tables.
7. The resolution of Figure 4 should be enhanced.
8. Sentence: "In this real case, we obtained a total of 20 batches of data through fermentation. of which 15 batches were used as training sets and 5 batches were used as test sets." There should be a comma in the middle of the sentence, not a full stop.
9. Tables: R2, symbol "2" should be the upper style.
10. The authors can give complexity analysis or show the running times of all comparison methods.
11. Conclusion is very simple. Conclusion should point out the main findings of this research and analyze the shortcomings of the paper, and further indicate the future research direction.
Reviewer 2 Report
Comments and Suggestions for Authors
The article «Collaborative Multiple Players to Address Label Sparsity in Quality Prediction of Batch Processes» is devoted to a very important topic, a technique, called co-training, for leveraging only a small ratio of labeled data, to hone and formulate a more advantageous framework in soft sensor modeling.
The relevance of the work is determined by for decades, soft sensor has been extensively renowned for its efficiency in real-time tracking of expensive variables for advanced process control. However, despite with diverse efforts lavished on enhancing their models, the issue of label sparsity when modeling the soft sensors has always posed challenge across various processes.
The author's approach is different in that dissimilar to the conventional routine where only two players are employed, were investigated the efficient number of players in batch processes, making a multiple player learning scheme to assuage the sparsity issue
It is especially worth highlighting the conclusion to which authors comes that the forged framework can outperform the other prevalent methods especially when the ratio of unlabeled data is climbing-up, and two case studies are showcased to demonstrate its effectiveness.
The article presents an original and complete study, the conclusions of which are very important for the current stage of development of research.
Comments and Suggestions for Authors:
Generally the conclusions consistent with the evidence and arguments presented.
However, in order to make the results more understandable to the reader, it is proposed to correct the following provisions:
· Please describe in more detail the data on which the study was conducted.
· Please explain why the penicillin fermentation simu- 258 lation platform PenSim V2.0 to generate simulated data was selected.
· For theoretical framework and bibliography additional current references should be included to new researches 2023 – 2024.
· Please adjust the structure of the article according to the requirements
· The process of discussing the results can be e extended by applying the results and extrapolating them to other similar studies.
· Please describe in detail how your study fits for aims and scope of Sensors
The comments are of a clarifying nature and in no way diminish the great merit of the author who proposed the new original method.
Reviewer 3 Report
Comments and Suggestions for Authors
This manuscript proposes a semi-supervised deep learning model called CO-MP-DSSM to address label sparsity in soft sensor modeling of batch processes. While the goal is worthwhile, there are several technical issues and areas for improvement:
(1) The introduction lacks a strong motivation. While it mentions challenges in soft sensor modeling, a more compelling argument is required to explain why current methods are insufficient and why this approach outperforms existing related work.
(2) The model architecture lacks clarity in its explanation. Several crucial aspects of CO-MP-DSSM are either overlooked or inadequately detailed:
Ÿ The rationale behind the number and types of base regressors utilized within the collaborative multiple players structure remains unsubstantiated. Further analysis is essential to ascertain the optimal quantity and to provide justification for how distinct regressor types may capture complementary features.
Ÿ Insufficient elaboration is provided on the 2D sliding window technique. Additional information is warranted on the process of selecting window sizes, reducing feature dimensionality post-sliding, and the manner in which dynamics across both dimensions are addressed.
Ÿ The description of how pseudo labels from different regressors are amalgamated is ambiguous. Clarifying the aggregation method and its impact on model performance would enhance the efficacy of the approach.
Ÿ More comprehensive details pertaining to the deep synthesizer GSTAE are necessary, including insights into layer structures, the optimization process, and the integration of quality information at various layers.
(3) Theoretical analysis supporting design choices is limited. Although the framework intuitively integrates well-known techniques, a theoretical foundation illustrating the advantages of this approach is absent. Comparative analyses of various architectural variants would assist in justifying the significant contributions made.
(4) The experimental evaluation lacks rigor and presents several shortcomings:
Ÿ Hyperparameters, including window sizes and model structures, are chosen without rigorous tuning procedures, potentially leading to overfitting the methods to the specific case studies.
Ÿ It is recommended to include additional performance metrics beyond RMSE/R2, such as assessing robustness to labeling noise and providing uncertainty estimates.
Ÿ The absence of statistical significance testing in the results makes it unclear whether improvements are statistically significant or merely a result of randomness.
(5) The organization and flow of the writing could be enhanced:
Ÿ Enhance readability by clearly segregating related work, methodology description, and evaluation.
Ÿ Improve clarity by structuring subsections and paragraphs to clearly delineate different technical components.
Ÿ Ensure consistency by defining jargon and notation to avoid ambiguity.
In conclusion, although the proposed model concept is well-founded, significant technical gaps, theoretical weaknesses, and limitations in the evaluation indicate that substantial revisions are necessary before the approach and conclusions can be deemed robust.
Round 2
Reviewer 1 Report
Comments and Suggestions for Authors
The authors have addressed all my concerns.
Reviewer 3 Report
Comments and Suggestions for Authors
The modifications made in the paper make it suitable for publication.